# Cell-Death Metabolites from *Cocconeis scutellum* var. *parva* Identified by Integrating Bioactivity-Based Fractionation and Non-Targeted Metabolomic Approaches

**DOI:** 10.3390/md22070320

**Published:** 2024-07-18

**Authors:** Carlos Sanchez-Arcos, Mirko Mutalipassi, Valerio Zupo, Eric von Elert

**Affiliations:** 1Institute for Zoology, Cologne Biocenter University of Cologne, 50674 Köln, Germany; 2Department of Integrative Marine Ecology, Stazione Zoologica Anton Dohrn, 80122 Napoli, Italy; mirko.mutalipassi@szn.it; 3NBFC, National Biodiversity Future Center, 90133 Palermo, Italy; 4Integrative Marine Ecology Department, Stazione Zoologica Anton Dohrn, Ischia Marine Center, Punta San Pietro, 80077 Ischia, Italy

**Keywords:** *Cocconeis scutellum* var. *parva*, *Hippolyte inermis*, metabolomics, bioactivity-based fractionation, cell-death metabolites, LC-MS, diatoms

## Abstract

Epiphytic diatoms growing in Mediterranean seagrass meadows, particularly those of the genus *Cocconeis*, are abundant and ecologically significant, even in naturally acidified environments. One intriguing aspect of some benthic diatoms is their production of an unidentified cell-death-promoting compound, which induces destruction of the androgenic gland in *Hippolyte inermis* Leach, 1816, a shrimp exhibiting protandric hermaphroditism, principally under normal environmental pH levels. The consumption of *Cocconeis* spp. by this shrimp is vital for maintaining the stability of its natural populations. Although many attempts have been made to reveal the identity of the apoptotic compound, it is still unknown. In this study, we strategically integrated a bioactivity-based fractionation, a metabolomic approach, and two different experimental avenues to identify potential apoptotic metabolites from *Cocconeis scutellum* var. *parva* responsible for the sex reversal in *H. inermis*. Our integrated analysis uncovered two potential candidate metabolites, one putatively identified as a lysophosphatidylglycerol (LPG) (16:1) and the other classified as a fatty acid ester. This is the first time LPG (16:1) has been reported in *C. scutellum* var. *parva* and associated with cell-death processes. These candidate metabolites mark substantial progress in elucidating the factors responsible for triggering the removal of the androgenic gland in the early post-larval phases of *H. inermis*.

## 1. Introduction

Diatoms serve multifaceted roles within both planktonic and benthic ecosystems [1], producing a diverse array of bioactive compounds [2] that significantly impact the physiology and reproductive patterns of planktonic crustaceans [3,4]. Despite this, the functional significance of benthic diatoms remains relatively underexplored [5,6]. Nonetheless, recent decades have witnessed a surge in research into the physiological effects of benthic diatoms [7]. Notably, studies have revealed the profound influence of *Cocconeis* diatoms on the physiology and ecology of the shrimp species *Hippolyte inermis* Leach, 1816 [8]. These investigations have unveiled how seasonal diatom consumption triggers early reversal from male to female sex in these shrimp [9]. Furthermore, more recent studies have demonstrated that diatom ingestion prompts rapid degeneration of the shrimp’s androgenic gland (AG) [10], with this process being linked to ferroptosis-induced cell death within the AG [11].

It is widely acknowledged that in crustaceans, the male sex relies on the presence of a specific gland known as the AG, and the activation of a specific trigger, termed the insulin-like androgenic gland (IAG), is adequate to maintain the male sex phenotype. In the absence of the IAG, crustaceans naturally transition to the female sex [12]. Consequently, the activity of diatoms, which destroy the AG in the early phases of development (though after a few days of post-larval growth) [11], is adequate to induce a rapid shift toward femaleness.

The quest to identify the compound or group of compounds from *Cocconeis scutellum* var. *parva* (Grunow) Cleve, 1895 responsible for the sex reversal in *H. inermis* began over a decade ago with the evaluation of three preparative-based fractions obtained from the diethyl ether extract of *C. scutellum* [13]. A GC-MS analysis of the fractions suggested the potential involvement of eicosapentaenoic acid (EPA) in apoptotic activity. Then, a bioassay-guided fractionation approach, including various sub-fractionation processes, revealed the presence of a highly lipophilic compound [14], the identity of which remains elusive despite these efforts.

Recently, Mutalipassi et al. (2019) [15] revealed that diatoms grown in artificial ocean-acidified conditions generated a proportion of females in *H. inermis* that was significantly lower than the diatoms produced at a normal pH. Those results indicate that *C. scutellum* grown under normal pH conditions (8.2) exhibits a higher sex reversal activity on *H. inermis*, implying that the responsible infochemical may also be more abundant under these conditions. Other studies have shown alterations in the exometabolome of *C. scutellum* under acidified conditions, affecting recognition by *H. inermis* [16]. However, these studies have largely overlooked exploration of the metabolic profiles of *C. scutellum* in search of the sex reversal infochemical.

Metabolomics, particularly non-targeted MS-based methods, has significantly advanced our comprehension of the intricate chemical interactions shaping ecological systems [17]. Beyond merely capturing snapshots of metabolites, advancements in instrumentation have empowered researchers to explore more deeply the molecular dialogues within ecosystems. Moreover, the evolution of open-source software for metabolomic data analysis has democratized access to powerful analytical tools, streamlining the extraction of meaningful insights from complex datasets [18,19]. Thus, recognizing the persistent challenge of identifying the specific infochemicals from *C. scutellum* var. *parva* responsible for the sex reversal phenomenon in the shrimp *H. inermis*, this study strategically integrates non-targeted MS-based metabolomics with traditional bioactivity-based fractionation approaches. Through a combination of bioactivity assays and comparative metabolomics analyses of *C. scutellum* var. *parva* grown under different pH conditions, our objective is to unravel and confirm the presence of metabolites in *C. scutellum* var. *parva* potentially involved in the sex reversal events observed in the shrimp *H. inermis*.

## 2. Results

### 2.1. Interaction of Sex-Reversal Activity from C. scutellum parva with Anion-Exchanger

When *Cocconeis scutellum* var. *parva* extract was passed through a strong anion-exchanger (‘SAX’), the run-through exhibited low activity in terms of the sex reversal, with a female/mature value of 24 ± 12% (Appendix A), which was not different from the negative control (*Z*-test, *p >* 0.05). However, the fraction that had bound to this anion-exchanger proved to cause sex reversal in the shrimp *Hippolyte inermis* (females/mature value of 58 ± 13%, Appendix A), and a *Z*-test indicated that this activity was significantly higher than in the negative control (*p* < 0.001). This revealed that the sex-reversing infochemical was negatively charged. When *C. scutellum* var. *parva* extract and the fraction ‘bound to SAX’ were compared by HPLC, it became evident that binding to and subsequent elution from this anion-exchanger caused a substantial removal of biologically non-active components from the *C. scutellum* var. *parva* extract (Appendix A)

### 2.2. Fractionation of Sex-Reversal Activity from C. scutellum var. parva by HPLC

When *C. scutellum* var. *parva* extract was subjected to HPLC after pre-cleaning by anion-exchanger, eight fractions were collected, covering the whole chromatogram and, hence, the whole spectrum, from hydrophilic to lipophilic components. The highest activity in terms of sex reversal was measured in fraction 2, with a F/mat% value of 81 ± 11% (Figure 1). A *Z*-test indicated that this fraction was significantly more active than the negative control (*p* < 0.05). The sex-reversal activity was confined to this single fraction, as the activity of none of the other HPLC fractions was significantly different from the negative control (Figure 1).

### 2.3. Molecular Features Present in Active Fraction F2 and Anion-Exchanger Extract

To establish candidate metabolites responsible for inducing the sex reversal of *Hippolyte inermis*, a pattern search was executed on the metabolic profiles of the fractions collected from the *C. scutellum* var. *parva* cells extract. Three features were identified, correlating with the criteria of being present in higher levels in the active fraction F2 and the SAX extract compared to the non-active fractions (Table 1).

### 2.4. Metabolic Profiles of C. scutellum parva Differ When Cultivated under Different pH Conditions

A PLS-DA was carried out to determine if the metabolic profiles of *C. scutellum* var. *parva* cells cultivated under the two pH conditions can be discriminated. The PLS-DA plot reveals discrimination between the metabolic profiles of algae cultivated at pH 7.6 and those at pH 8.2. Component 1 explains 29% of the variability of the data and separates the metabolic profiles of *C. scutellum* var. *parva* cultivated at pH 7.6 on the left from those at pH 8.2 on the right. Component 2 explains 25.2% of the total variability of the data but does not separate the profiles (Figure 2).

### 2.5. Active Molecular Features Up-Modulated under pH 8.2

Considering that in previous studies, the sex reversal of *Hippolyte inermis* was observed when feeding on *C. scutellum* var. *parva* cultured under conditions of pH 8.2 [15], a volcano plot comparing the metabolic profiles of pH 8.2/7.6 was executed to identify features that were up-modulated under high pH conditions (Figure 3A). A total of 13 molecular features (each representing a different metabolite) were identified as being present in significantly higher amounts ((FC) ≥ 2.0 (*p* ≤ 0.05)) in *C. scutellum* var. *parva* cells cultivated at 8.2 compared to those cultivated at 7.6 (red dots, Figure 3A). Among all the up-modulated features, we were able to identify the presence of the active molecular features ID 295 (same as ID 2126 in Table 1) (Figure 3B) and ID 1069 (same as ID 2470 in Table 1) (Figure 3C). This confirms that the levels of the two active molecular features previously identified in the active fraction F2 and SAX extract (Section 2.2) are also up-modulated at pH 8.2.

Feature ID 295 (same as feature ID 2126 in Table 1), representing a compound with *m*/*z* of 481.257 (Table 2), was putatively identified as a lysophosphatidylglycerol: 1-(9Z-hexadecenoyl)-glycero-3-phospho-(1′-sn-glycerol) (LPG (16:1), SIRIUS score: 91.078%, total explained intensity: 90.628%). The fragmentation pattern from this compound is depicted in Figure 4. Conversely, the feature identified as ID 1069 (same as feature ID 2470 in Table 1) did not display a high-score match over the databases. Still, it was classified as a fatty acid ester by the compound class prediction tool (CANOPUS) [20,21] from the SIRIUS toolbox [22].

## 3. Discussion

Our study employed a strategic integrative approach, combining two distinct analytical tools and experimental avenues, to uncover and confirm potential metabolites from *Cocconeis scutellum* var. *parva* that could drive the sex reversal phenomenon in the shrimp *Hippolyte inermis*. This comprehensive methodology allowed us to identify two promising metabolites, one of which was putatively identified and might be responsible for triggering the development of beta females in *H. inermis*. These metabolites represent a significant advance in our quest to pinpoint the apoptotic metabolic factor that triggers the rapid degradation of the androgenic gland during the early post-larval stages of *H. inermis*.

Our first avenue of investigation involved integrating bioassay-guided fractionation with metabolomic analysis, leading to the identification of higher levels of three candidate molecular features (each representing a different metabolite) in fraction F2 and in the anion-exchange extract (Table 1 and Appendix A) from *C. scutellum* var. *parva* cells. Notably, these molecular features correlated with a higher percentage of induced females, highlighting the efficacy of our integrated approach in prioritizing biologically active fractions or extracts for further analysis. In previous attempts [14], two active fractions, ‘hydrophilic and lipophilic’, have been described from the same diatoms. Considering that they were obtained under different chromatographic gradients and mobile phases (solvent A: 80/20, *v*/*v*, MeOH/H_2_O and solvent B: 80/20, *v*/*v*, MeOH/acetone), we could speculate that our active F2 fraction and their ‘hydrophilic’ fraction can share metabolites with related chemical characteristics because both of them eluted under low concentrations of the respective organic modifiers.

In our second avenue, inspired by previous findings that ocean acidification affects the volatile profiles of *C. scutellum* var. *parva* [16], we compared the endometabolic profiles of diatoms grown under normal (pH 8.2) and acidified (pH 7.6) conditions. Our results showed distinct changes, indicating that ocean acidification influences the endometabolome of *C. scutellum* var. *parva*. Additionally, considering evidence that suggests that *C. scutellum* var. *parva* grown under normal pH conditions leads to higher sex reversal in *H. inermis* [15], we identified 13 up-modulated metabolites under a normal pH (8.2) through fold-change analysis. Notably, two molecular features (ID 295 and 1069, representing compounds with *m*/*z* 481.257 and 497.3119) corresponded to metabolites highly present in active fractions (feature IDs 2126 and 2470, Table 1). These findings highlight the potential of these two metabolites as candidates for inducing sex reversal in *H. inermis*, as they are significantly more abundant under pH 8.2, where the proportion of females in *H. inermis* is higher.

The metabolite corresponding to molecular feature ID 295 (or ID 2126 in Table 1), with a *m*/*z* 481.257, was putatively identified as a lysophosphatidylglycerol: 1-(9Z-hexadecenoyl)-glycero-3-phospho-(1′-sn-glycerol), also known as LPG (16:1), and represents a novel finding in *C. scutellum* var. *parva*. LPG (16:1) has also been identified in various microalgae, like *Chlamydomonas reinhardtii*, *Chlorella vulgaria*, *Nannochloropsis* sp., *Scenedesmus* sp., and *Schizochytrium limacinum* [23], as well as in several macroalgae, like *Ulva rigida* [24], *Palmaria palmata* [25], *Grateloupia turuturu* Yamada [26], *Porphyra dioica* [27], and *Undaria pinnatifida* [28]. While experimental evidence concerning the apoptotic activity of LPG (16:1) is lacking, lysophosphatidylglycerols (LPGs) are a subclass of phospholipids known for their role in facilitating apoptosis [29]. It is only understood that their mechanism of action involves the induction of apoptosis, potentially through the engagement of proteins within the death receptor pathway, subsequently triggering the activation of the intrinsic apoptotic pathway [30]. Thus, our results are the first that suggest the potential role of LPG (16:1) as an apoptotic factor involved in the sex reversal of *H. inermis*.

Similarly, LPG is a lysophospholipid subclass involved in various cellular functions [31]. Although their precise roles in apoptosis are not fully elucidated, some studies suggest potential mechanisms through which they may influence apoptotic processes. LPG may disrupt cellular membranes, potentially affecting membrane integrity and function. Disruption of membrane integrity can trigger apoptotic signaling pathways [32]. LPG could also modulate intracellular signaling pathways involved in apoptosis, acting through G-protein-coupled receptors [33]. In addition, LPGs might induce changes in the composition of the mitochondrial membrane; for instance, regulating mitochondrial Ca^2+^ transport [34], impacting mitochondrial permeability, and potentially releasing apoptotic factors. Additionally, LPGs can be detected by lysophospholipid receptors in immune cells, modulating immune responses and promoting inflammatory processes [35]. As inflammatory responses can intersect with apoptotic pathways [36], LPGs may contribute to apoptosis indirectly through their effects on inflammation.

Our findings indicate that LPG (16:1) is more abundant in *C. scutellum* var. *parva* under normal pH conditions (pH 8.2) than under acidified conditions (pH 7.6) (Figure 3B). Similarly, changes in the LPG (16:1) content have also been reported in the brown algae *Fucus vesiculosus*, but due to seasonality [37]. Conversely, the levels of other lysophospholipid classes are up-modulated in *Phaeodactylum tricornutum* [38], *Ulva prolifera* [39], and *Galdieria sulphuraria* [40] under ocean acidification conditions. Furthermore, lysophospholipids also endure seasonal plasticity in the macroalgae *Ulva rigida* [41] and depletion under nitrogen starvation of the diatom *Phaeodactylum tricornutum* [42]. The modulation of these lysophospholipids can be related to their widely accepted role as signaling molecules and mitigators during plants’ stress responses [43] to environmental cues [44], abiotic stress [45], and physical damage [46]. Thus, the content changes of LPG (16:1) observed in our study might indicate how phenotypically plastic *C. scutellum parva* cells respond to variations in ocean CO_2_ levels. Ocean acidification will affect endometabolites and strongly impact chemical communication [47,48]. Within a future ocean and due to the potential role of LPG (16:1) as an infochemical, these endometabolomic responses of *C. scutellum* var. *parva* could most probably be propagated to the level of its grazer *H. inermis*, where the reduced levels of LPG (16:1) may lead to lower rates of sex reversal and hence lower frequencies of females in *H. inermis* populations.

The second metabolite corresponding to the selected molecular feature ID 1069 (or ID 2470 in Table 1), with *m*/*z* 497.3119, was classified as a fatty acid ester. Still, no strong match was found in the databases. Zupo et al. (2014) suggested that the apoptotic metabolite from *C. scutellum parva* could belong to the lipid subclass of free fatty acids. In line with this assumption, a polyunsaturated fatty acid (PUFA): dihomo-γ-linolenic acid (DGLA; 20:3n-6) displayed high sex reversal activity in *H. inermis* [11]. While DGLA was present in the *C. scutellum* var. *parva* anionic-exchanger and whole acetonitrile extracts in our study, it was detected toward the hydrophobic session of the chromatograms belonging to fractions that were not significantly active (Appendix A). Furthermore, fatty acid derivatives, like aldehydes [49] and oxylipins [50], have previously been identified as metabolites involved in diatom–copepods interactions. Our findings and the previous evidence of fatty acid-derived metabolites involved in the interactions of diatoms with other species indicate the potential of this still-unknown metabolite to be involved in the diatom–grazer interaction in our study. Further research is needed to evaluate the potential and identify this metabolite

The integration of bioactivity-based fractionation and metabolomics provided significant insights into the potential apoptotic metabolites from *Cocconeis scutellum* var. *parva* involved in sex reversal in *Hippolyte inermis* and proved advantageous over classical bioassay-guided fractionation methods. This holistic strategy enabled a more comprehensive analysis of the complex chemical environment and facilitated the identification of potential bioactive compounds with greater specificity and efficiency. The chromatographic techniques and mobile phases selected were particularly effective in resolving these potential compounds, underscoring the importance of optimal methodological choices in such studies. Our LC-MS analysis revealed two candidate metabolites: one putatively identified as a lysophosphatidylglycerol (LPG 16:1) and another classified as a fatty acid ester. While LPG (16:1) is newly reported in *C. scutellum* var. *parva* and associated with apoptosis, further research is required to validate its involvement in the sex reversal phenomenon and fully understand its specific role in the apoptotic process. Additionally, the fatty acid ester, although not fully annotated, contributes to the understanding of the biochemical triggers of androgenic gland removal in *H. inermis*. Investigating the metabolite’s receptors, synthetic enzymes, and other pathophysiological roles could offer valuable insights into their broader cellular functions.

## 4. Materials and Methods

### 4.1. Collection and Isolation of the Diatom Cocconeis scutellum var. parva

*Cocconeis scutellum* var. *parva* are routinely cultured at the Benthic Ecology Laboratory of the Stazione Zoologica Anton Dohrn in Ischia. The strains were collected in 2006 from the Bay of Naples, Italy (40.44.7 N, 13.56.4 E), from a *Posidonia oceanica* meadow located in Lacco Ameno d’Ischia (Island of Ischia), isolated by a Leica micromanipulator, cleaned of bacteria and other diatoms through several passages in sterilized seawater, and then identified under the SEM microscope. Stock cultures were held in plastic multi-well plates and transferred biweekly into a new sterile medium (Guillard’s f/2, Sigma Aldrich, Steinheim, Germany) [51] using autoclaved Pasteur pipettes. A sample of each strain was collected from the stock cultures and multiplied in (7 cm diameter) glass Petri dishes before starting the experiments.

### 4.2. Collection of Hippolyte inermis and Larval Production

Ovigerous females of *Hippolyte inermis* were collected from a seagrass meadow off Lacco Ameno d’Ischia (Island of Ischia, Naples, Italy) by horizontally towing a plankton trawl over the *P. oceanica* canopy. *H. inermis* specimens were sorted manually onboard by well-trained operators to separate them from bycatch. Fortunately, *H. Inermis* specimens can be easily identified in the Ischia meadows as no similar species are present, ensuring accurate sorting. They were temporarily stored in plastic bags until transportation to the laboratory. Taxonomical identification was confirmed under the macroscope (Leica Z16-APO macroscope, Wetzlar, Germany). Each ovigerous female was then individually relocated into 2 L conical flasks containing approximately 1.8 L of filtered seawater and a small portion of *Posidonia* leaf for shelter. These flasks were placed in a thermostatic chamber set at 18 °C (based on the average seawater temperature at the field sites during spring) [52,53], illuminated with Gro-Lux fluorescent tubes providing a mean irradiance of 250 μmol m^−2^ s^−1^ for ten hours daily. After a variable period of a few days, each female released larvae (ranging from 20 to 400), which were promptly collected for subsequent bioassays [8,52], while the mothers were released back into the sea. The collected larvae were grouped into batches of 80 and transferred to 1 L conical flasks containing approximately 800 mL of filtered and sterilized natural seawater, still within the same thermostatic chamber described earlier.

### 4.3. Bioactivity-Guided Fractionation and Identification of Molecular Features from LC-MS Data

#### 4.3.1. *C. scutellum* var. *parva* Culture and Harvest of Biomass for Fractionations

All the operations further described were performed under a laminar flow hood, and the dishes and all the laboratory instruments were previously autoclaved at 120 °C. The cells growing on 4 cm wells (mother cultures, as previously described) were scraped and collected by a Pasteur pipette 15 days after inoculation. The suspensions derived from 6 wells were pooled into a sterilized beaker and divided into 20 aliquots to be transferred to 20 (7 cm diameter) Petri dishes filled with 200 mL f/2 medium. Dense diatom biofilms were observed on the Petri bottoms after 15 days of growth in the thermostatic chamber under Gro-Lux fluorescent light (about 150 µE irradiance) with a 12/12 h photoperiod. They were scraped with a sterile Pasteur pipette, collected and pooled in a sterilized beaker. Then, 2 mL of this denser suspension was inoculated into each of 200 Petri dishes (14 cm diameter) containing 150 mL of f/2 medium. After the inoculation, the diatoms were cultured for 15 days in the same thermostatic chamber under conditions identical to those previously described. Finally, after removal of the medium, the dishes were quickly rinsed twice with distilled water, frozen at −20 °C and lyophilized overnight. The dried diatoms were scraped off with a blade, weighed and kept in dry glass vessels at −20 °C. Since each Petri dish produced, in 15 days of culture, about 5 mg of dried weight (DW) of diatoms, on average, this procedure was repeated ten times to obtain 10 g DW of diatoms, which was homogenized entirely with a spatula before starting the analytical practices.

#### 4.3.2. Extraction, Purification and Fractionation of *C. scutellum* var. *parva*

##### Extraction

As previously reported in Zupo et al. (2014), 1 g of freeze-dried *Cocconeis scutellum* var. *parva* powder was suspended in 10 mL of ultrapure water, incubated at room temperature for 10 min, and then exposed to ultrasonication for 3 min. Then, 80 mL of acetonitrile was added, the suspension was stirred for 3.5 h in the dark at room temperature and then stored overnight in the dark at 4 °C. Then, the suspension was stirred for another two hours in the dark at room temperature. Then, the particles were removed by centrifugation (5 min, 17,000 *g*). The clear supernatant was removed, filled to 100 mL using 80% (*v*/*v*) acetonitrile, and then 150 mL of ultrapure water were added to generate extract A.

##### Lipophilic Solid-Phase Extraction

A C_18_ cartridge (Mega Bond Elut-C18, 10 g, Agilent, Waldbronn, Germany) was activated with 50 mL acetonitrile and conditioned with 50 mL 30% acetonitrile. Then, extract A was passed through the cartridge, which was subsequently washed first with 25 mL of 30% acetonitrile and then with 50 mL of 60% acetonitrile. The sample of interest was collected by eluting the cartridge with 100 mL each of 80% acetonitrile and then 100% acetonitrile. Both eluates were pooled and evaporated to dryness (rotatory evaporator, 40 °C and speedvac) and re-dissolved in 1 mL acetonitrile (extract B).

##### Anion-Exchange Solid-Phase Extraction

In order to identify another property of the biologically active substance in addition to lipophilicity that can be used for chemical characterization, we tested for a negative charge of the substance using an anion-exchanger. To 900 µL of extract B, 2.1 mL of acetonitrile were added, and then 27 mL of ultrapure water were added to yield 30 mL of extract C. A strong anion-exchanger cartridge (SAX, Agilent 12102044, 500 mg, 3 mL) was rinsed with 5 mL of ultrapure water, and the run-through was discarded. Then, 10 mL of extract C and subsequently 2 mL of ultrapure water were passed through the SAX cartridge and collected, and 5 mL of acetonitrile was added (‘SAX run-through’). Compounds bound to the anion-exchanger were eluted by rinsing the cartridge with 10 mL of 1 M NaCl solution, to which 50 mL of ultrapure water and 15 mL of acetonitrile were added (‘SAX-NaCl’). The same procedure was repeated with the two remaining 10 mL aliquots of extract C, each using a new SAX cartridge. The three samples of ‘SAX run-through’ were pooled, and so were the three samples of ‘SAX-NaCl’. Each of these two samples, ‘SAX run-through’ and ‘SAX-NaCl’, were then subjected to lipophilic solid-phase extraction (see Section Lipophilic Solid-Phase Extraction) in order to re-extract the active compounds and thereby remove the accompanying salts, which yielded 900 µL each of ‘SAX run-through extract’ and of ‘SAX-NaCl extract’.

##### Chromatography and Fractionation by HPLC

For the fractionation, aliquots of 30 µL were injected into a Shimadzu LC-20AB HPLC (Kyoto, Japan) equipped with a SIL-20AS autosampler, a SPD-M20A diode array detector, a CTO-10AC column oven and a FRC-10A fraction collector. Chromatography was performed on a C_18_ reverse-phase Grom-Sil 120 ODS-4 HE, 5 µm column (250 × 4.6 mm) with ultrapure water (solvent A) and acetonitrile (solvent B) as mobile phases. The following chromatographic method was applied: 0–10 min isocratic 40% B, 10.1–15 min gradient phase to 70% B, 15.1–40 min gradient to 100% B, 40.1–45 min isocratic 100% B, 45.1–46 min gradient to 40% B, and 46.1–50 min isocratic 40% B. Identical fractions collected from multiple runs were subsequently pooled, evaporated to dryness, and shipped to the Ischia lab for bioassay.

#### 4.3.3. Bioassay with *Hippolyte inermis*

The larvae were nourished for the initial seven days with a diet comprising four *Artemia salina* nauplii and four *Brachionus plicatilis* individuals per mL of seawater. Subsequently, the administration of *Brachionus* was halted, and *Artemia nauplii* was enriched using an Algamac, Biomarine (Hawthorne, CA, USA) integrator. After approximately 30 days, when the larvae settled, they were moved in batches of 25 post-larvae to vessels measuring 14 cm in diameter, each containing 400 mL of filtered seawater. These post-larvae were fed a composite diet consisting of three items in equal proportions: SHG ‘Artemia Enriched’, SHG ‘Microperle’, and SHG ‘Pure Spirulina’, which constituted the base food for the negative controls. To assess the activity of *C. scutellum* var. *parva*, dried fractions from the Section Chromatography and Fractionation by HPLC were re-dissolved in 2 mL of methanol (MeOH) and incorporated into the prepared composite food. The solvent was then evaporated using a rotary evaporator to yield dried foods supplemented with the diatom fractions.

For the positive controls (‘Plus’), dried diatoms (*C. scutellum* var. *parva* from identical batches) were added to the base food at 2:1 ratio (*w*/*w*) of base food/diatoms, respectively. All the food preparations were completed at the onset of the experiment (the first day of post-larval growth) and stored in a freezer at −20 °C until administration. Daily, 5 mg of dry food, or dry food combined with diatoms, was administered to each vessel containing 25 *H. inermis* post-larvae, according to the specific treatment.

Specifically, we conducted three replicates (25 individuals each) for the negative and positive controls and three replicates (25 individuals) for each dried fraction obtained from HPLC (Section Chromatography and Fractionation by HPLC) under examination. Mortality was monitored daily (dead larvae are pale white without movement compared to the copper color of alive and mobile larvae) by collecting and counting larvae using a Pasteur pipette, regularly replacing the water and food. The post-larvae were euthanized after 40 days, preserved in 70% ethanol, and subsequently examined under a dissecting microscope to measure total body length using millimeter paper and a metal bar. Their second pleopods were mounted on slides and examined under an optical microscope, as described in Mutalipassi et al. (2018) and Zupo et al. (2008), [54,55] to determine the presence or absence of a masculine appendix, indicating male or female sex, respectively.

Data were expressed as the percent females per mature animals. The square root of these values was arc-sin transformed and used for statistics. *Z*-tests were run for pairwise comparison of the activity of fractions with those of the negative controls.

#### 4.3.4. Acquisition of Fractions LC-MS Metabolic Profiles

The LC-MS metabolic profiles of the fractions obtained in the Section Chromatography and Fractionation by HPLC were acquired by injecting three technical replicates of 10 µL of each sample into a Vanquish™ Horizon UHPLC system coupled to a Q-Exactive HF mass spectrometer (Thermo Fisher Scientific, Dreieich, Germany). Separation of the metabolites was performed at a constant flow rate of 300 µL min^−1^ on an EC 125/2 Nucleosil 100-3 C_18_ column (Macherey-Nagel, Düren, Germany) set at 30 °C, by using a gradient of 0.1% (*v*/*v*) formic acid in water (solvent A) and 0.1% formic acid in acetonitrile (solvent B) as follows: 0.0–0.3 min isocratic 20% B, 0.3–7.0 min gradient phase to 100% B, 7.0–11.0 min isocratic 100% B, 11.0–11.1 min gradient to 10% B, 11.1–13.0 min isocratic 20% B. The samples were measured in the negative ionization (NI) mode (considering the negatively charged nature of the metabolites’ product of the strong anion-exchanger solid-phase extraction from the Section Anion-Exchange Solid-Phase Extraction), with a resolving power of 120,000 m/Δm and a mass range of 75–1125 *m*/*z*. The electrospray ionization (ESI) source was set at 3.0 kV (NI) and 3.7 kV (PI) spray voltage and a capillary temperature of 360 °C. Full MS scan and data-dependent MS2 (Full MS/dd-MS2 (top N)) acquisitions were performed on each sample. For the full MS acquisition, a resolution of 120,000 m/Δm, automated gain control (AGC target) was set to 1 × 10^6^, the scan range of 75–1125 *m*/*z*, and a maximum injection (IT) time of 100 ms were used. The parallel dd-MS2 (top N) was acquired for the top 20 most intense *m*/*z* by using a resolution of 120,000 m/Δm, an AGC target of 1 × 10^5^, max IT of 100 ms, an isolation window of 1.0 *m*/*z*, and a normalized collision energies of 30, 45, and 60 eV.

#### 4.3.5. LC-MS Data Preprocessing

All the Thermo raw data files obtained from the LC-MS acquisition were converted into .mzXML format utilizing the MSconvert tool from ProteoWizard 3.0x software [56]. The converted files of all the replicates were sorted into separate folders according to the fraction number or SAX extract, and then all the folders were processed together with the XCMS 3.14.1 R package [57,58,59]. Peak picking was performed by applying the centWave algorithm (ppm = 1, peakwidth = c(5, 30), prefilter = c(6, 50,000), mzCenterFun = ‘wMean’, integrate = 1, mzdiff = −0.001, fitgauss = TRUE, noise = 1 × 10^4^, verboseColumns = TRUE). The selected peaks were initially aligned (bw = 2, mzwid = 0.009, max = 100, minsamp = 0, sleep = 0). Then, the retention times were corrected with the obiwarp method, followed by a second alignment (bw = 2, mzwid = 0.009, max = 100, minfrac = 0.5, minsamp = 1, sleep = 0). All the masses detected in the blank and those with coefficients of variation for intensities higher than 20% were removed from the mass peak list. Finally, the CAMERA 1.48.0 package [60] was used for the isotopes’ and adducts’ annotation and to generate our final peak intensities table for further statistical analysis.

#### 4.3.6. Data Processing and Identification of Molecular Features from LC-MS Data

To identify features showing a particular pattern of change among the metabolic profiles of the different fractions actively inducing the sex reversal of *Hippolyte inermis* compared to the non-inducing fractions, we used the pattern hunter tool from MetaboAnalyst 3.0 [61]. The peak intensities table was initially square root-transformed and scaled by mean-centering and dividing by the square root of the standard deviation of each variable (equivalent to Pareto scaling). A Spearman rank correlation analysis was performed against the suggested patterns. Each pattern is a stated series of numbers, where each number indicates the expected change of the features among the metabolic profiles of the fractions. For example, the pattern ‘0-1-0-0-0-0-0-0-1’ corresponded to the levels in ‘F1-F2-F3-F4-F5-F6-F7-F8-SAX’ and was used to search for features more abundant in fraction F2 and SAX extract.

### 4.4. Non-Targeted Metabolomic Analysis of C. scutellum var. parva under Two Different pH Conditions

#### 4.4.1. *C. scutellum* var. *parva* Culture and Harvest of Biomass for Metabolomics

The cells cultivated in six-well mother-culture plastic dishes were harvested using a Pasteur pipette 15 days post-inoculation when diatoms nearly covered the surface. The collected diatoms were pooled into a sterilized beaker, and the suspension was divided into 20 portions and subsequently transferred to 20 Petri dishes (7 cm diameter) filled with f/2 medium. These dishes were incubated in a thermostat chamber (18 °C, photoperiod 12/12 provided by Gro-Lux neon lamps yielding an average PAR of 150 µE) for 15 days to cultivate a diatom biofilm. After 15 days, the surfaces of the Petri dishes were wholly coated with diatom biofilms, which were collected and pooled into a sterilized beaker. The diatoms were once more combined in a sterilized beaker, and the suspension was divided into eight photobioreactors (four for normal and four for acidified conditions), with each reactor filled with 2 liters of f/2 medium. Due to the adhesive nature of *Cocconeis* spp., only a fraction of the diatoms inoculated into each photobioreactor survived each reinoculation process. Consequently, the diatom concentration in the suspension was not determined at the time of inoculation.

#### 4.4.2. Set-Up of pH-Controlled Photobioreactors for Metabolomics

Special photobioreactors tailored to benthic diatoms were custom-designed to operate separately under normal (8.2) and acidified conditions (7.6), as described by Mutalipassi et al. (2019) [15]. Each photobioreactor was constructed using a Pyrex dish with a total volume of 2.4 L (300 mm × 200 mm × 40 mm; Figure 1). The vessel was covered with a heat-resistant glass plate featuring a narrow central opening for housing a pH probe (InLab Micro pH, Mettler Toledo, Columbus, OH, USA). A secondary opening was also established on the side, accommodating a plastic stripette. The InLab Microprobe is engineered to function efficiently even in minimal water volumes and up to a thickness of 3 mm. The pH levels in both treatments (pH 8.2 and pH 7.6) were controlled by a pH controller (pH 201, Aqualight, Salinas, CA, USA), linked to the InLab Micro Probe (via a BNC cable) through an electronic valve, which in turn was connected to a CO_2_ regulator (CO_2_ Energy, Ferplast, Castelgomberto, Italy). A centrifuge pump (Pure Pump 300, Askoll, Dueville, Italy) was integrated to prevent water stratification and the development of pH gradients within the photobioreactor. This centrifuge pump was regulated by a microcontroller (EnerGenie EG-PMS2-LAN), activating the pump cyclically every 30 min (1 min on, 29 min off).

The pH of the medium was monitored daily to ensure the pH oscillations remained below 0.05. Four replicates were generated for each pH condition. Diatom growth within the photobioreactors continued for 16 days. All the intact benthic diatom cells were firmly adhered to the bottom of the glass cups 24 h after inoculation. After 16 days, the medium in each photobioreactor was drained, and the vessels were swiftly washed in distilled water for 1 s to eliminate residual salts. Emptied vessels containing a diatom film on their bottom were promptly frozen at −20 °C and then subjected to freeze-drying. The dry diatoms were scraped off using an iron blade, weighed, and stored in dry vessels at −20 °C until used for metabolic extractions.

#### 4.4.3. Carbonate Measurements

Samples of the culture medium (50 mL in volume) were collected from each replicate every three days to analyze the seawater carbonate chemistry. Salinity was determined using a HI-96822 refractometer for seawater (Hanna Instruments, Woonsocket, RI, USA). The total alkalinity (TA) and pH (NBS scale; pHNBS) were measured using the total alkalinity mini titrator for water analysis HI-84531-02. Before each set of titrations, the electrode (pH 4.01, 7.01, and 8.30 at 25 °C) and pump (using a HI 84531-55 calibration standard) were calibrated to ensure accuracy. The instrument automatically adjusted to the working temperature of 18 °C using the automatic temperature compensation feature. Samples were analyzed immediately after collection from replicate photobioreactors, carefully avoiding the formation of air bubbles in the instrument that could affect the measurements. The seawater partial pressure of the CO_2_ analyses was determined using the CO_2_Sys EXCEL Macro [62,63,64] with pHNBS, TA, temperature, and salinity data. The carbonic acid dissociation constants (i.e., pK1 and pK2, [65]), ion HSO_4_ constant [66] and borate dissociation constant [67] were utilized for the computations.

#### 4.4.4. Endometabolome Extraction of *C. scutellum* var. *parva* Cells under Two Different pH Conditions

Ten mg of dried *Cocconeis scutellum* var. *parva* cells grown under normal (four replicates) and acidified (four replicates) conditions was ground separately in a tissue homogenizer using 1.4 mm zirconium oxide beads (Precellys, Montigny-le-Bretonneux, France). Then, 200 µL of prechilled Milli-Q water was added, vortexed vigorously for 30 s, and sonicated for one minute. After 30 min, 800 µL of acetonitrile (LC-MS grade) was added and vortexed again for 30 s. Samples were stored at 4 °C overnight to facilitate the extraction and then vortexed and centrifuged at 4000 *g* for 10 min. A total of 800 µL of supernatants was transferred separately into 1.5 mL glass vials and dried in a vacuum concentrator (RVC2-25, Christ, Osterode am Harz, Germany). Samples were finally re-dissolved in 300 µL of 100% acetonitrile (LC-MS grade) for LC-MS acquisition.

#### 4.4.5. Acquisition of LC-MS Profiles and Date Preprocessing

The LC-MS metabolic profiles of *Cocconeis scutellum* var. *parva* extracts grown under normal (pH 8.2) and acidified conditions (pH 7.6) were acquired, as described in Section 4.3.4. Similarly, data preprocessing was executed as described in Section 4.3.5.

#### 4.4.6. Data Processing and Statistical Analyses of LC-MS Metabolomics Data

Multivariate statistical analyses were performed to reveal the relationships between the metabolic profiles of *C. scutellum* var. *parva* cells cultivated at pH 7.6 and those at pH 8.2 using the MetaboAnalystR 3.0 package [61]. The peak intensities table was initially reduced by using the ‘getReducedPeaklist’ function within the CAMERA R package [60]. This function merges all the adducts of a potential compound into a single representative compound within each pcgroup (peak chromatogram group). As a result, the simplified peak list includes only one annotated feature per group. The reduced peak list was then normalized by sample-specific weights and later square root-transformed and scaled by mean-centering and dividing by the standard deviation of each variable (equivalent to auto-scaling). To visualize the changes between the metabolic profiles at pH 7.6 compared to 8.2, a partial least square discriminant analysis (PLS-DA) was carried out. To identify the most differentiating metabolites among the treatments, a volcano plot comparing the metabolic profiles of pH 8.2/pH 7.6 was executed (FC = 2.0, *p* < 0.05). Following this, molecular features selected from the bioactivity-guided fractionation experiment (identified in Section 4.3.6) were searched among the most differentiating metabolites.

#### 4.4.7. Compound Identification

Compounds with molecular features selected in the bioactivity-guided fractionation experiment and then present in the metabolomics experiment under the two different pH conditions were considered of high interest and putatively identified by calculating the exact masses of the metabolites from their accurate masses and the ion annotation information provided by CAMERA (adducts, isotopes, and neutral-losses fragments), executing an isotopic pattern analysis and performing library spectral matching allowing a mass deviation of 5 ppm in MetFrag and SIRIUS [68], using the Human Metabolome Database (HMDB), MassBank of North America (MoNA), KEGG, and LipidMaps databases.

## Figures and Tables

**Figure 1 marinedrugs-22-00320-f001:**
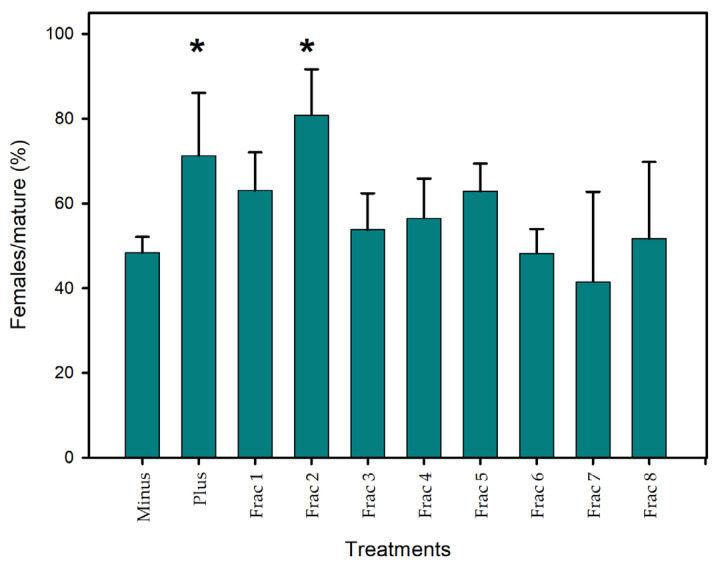
The sex-reversal activity of HPLC fractions of *C. scutellum* var. *parva* extract expressed as a percentage of females relative to the total number of mature animals. Depicted is the mean ± SE activity of *n* = 3 biological replicates of negative (‘Minus’), positive control (‘Plus’), and HPLC fractions 1–8. * indicates significant differences to ‘Minus’ after *Z*-test.

**Figure 2 marinedrugs-22-00320-f002:**
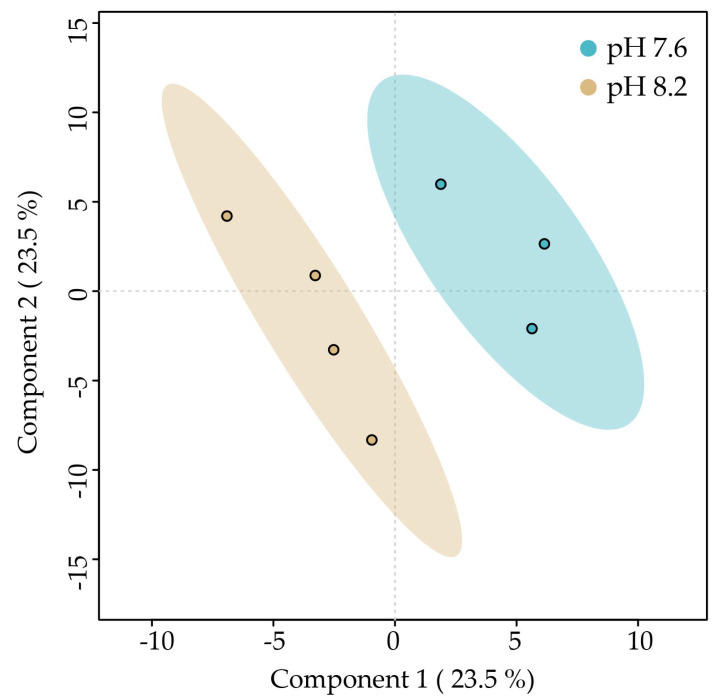
PLS-DA plot of the LC-MS-based metabolic profiles of *Cocconeis scutellum* var. *parva* cells cultivated at different pH levels. Small colored circles represent individual metabolic profiles of specimens cultivated at pH 7.6 (blue) and 8.2 (brown). Colored ellipses represent the 95% confidence regions for each group.

**Figure 3 marinedrugs-22-00320-f003:**
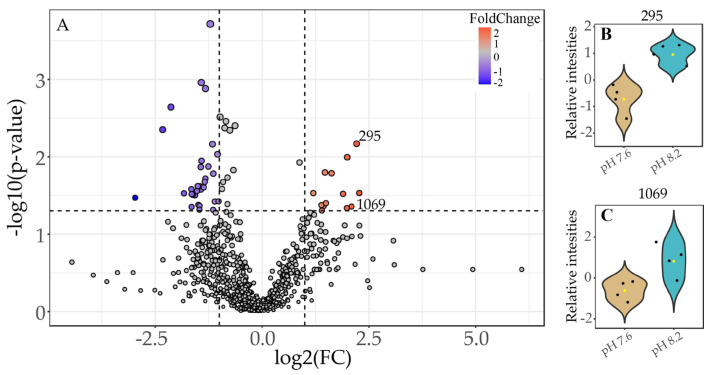
(**A**) Volcano plot of metabolites with fold changes (FCs) ≥ 2.0 (*p* ≤ 0.05) in *C. scutellum* var. *parva* (FC = amount at pH 8.2/amount at pH 7.6). The plot shows the −log 2 of the amount of each metabolite at pH 8.2 with respect to the amount of the same compound at pH 7.6. The red dots indicate metabolites with a FC ≥ 2.0 and *p* ≤ 0.05 higher at pH 8.2. The data presented correspond to the statistical analysis of four biological replicates for both pH values. On the right side are the violin plots of the relative intensities under different pH conditions of the molecular features (**B**) 295 and (**C**) 1069.

**Figure 4 marinedrugs-22-00320-f004:**
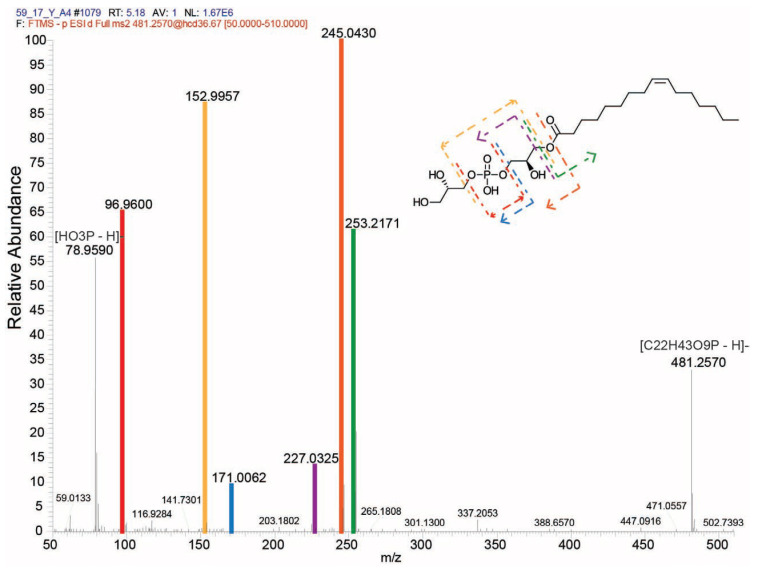
MS/MS fragmentation pattern of the compound represented by molecular feature ID 295. A lysophosphatidylglycerol putatively identified as 1-(9Z-hexadecenoyl)-glycero-3-phospho-(1′-sn-glycerol).

**Table 1 marinedrugs-22-00320-t001:** Features selected from pattern searching are highly present in F2 and SAX extract.

Feature ID	Representative Feature	[M-H]^−^ Precursor	Retention Time (min)	*p*-Value	FDR
2126	311.1866	481.2572	3.09	3.83 × 10**^−^**^10^	8.71 × 10**^−^**^8^
2412	334.1272	521.2883	5.09	5.57 × 10**^−^**^5^	4.07 × 10**^−^**^4^
2470	339.1999	497.2944	3.49	4.57 × 10**^−^**^4^	1.67 × 10**^−^**^3^

**Table 2 marinedrugs-22-00320-t002:** Feature IDs 295 and 1069 up-modulated under high pH conditions from the volcano plot of Figure 3 and also present in the active fraction F2 and SAX extract.

Feature ID	Fragment or Adduct	[M-H]^−^ Precursor	Retention Time (min)	*p*-Value	FC
295	311.1860	481.2570	5.18	0.0067882	4.638
1069	534.2884	497.3119	7.48	0.046058	3.957

## Data Availability

The data presented in this study are available on request from the **c**orresponding authors.

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
