# Peer review of "Cell-Death Metabolites from *Cocconeis scutellum* var. *parva* Identified by Integrating Bioactivity-Based Fractionation and Non-Targeted Metabolomic Approaches"

_marinedrugs, 2024, doi:10.3390/md22070320_

Round 1
Reviewer 1 Report
Comments and Suggestions for Authors
Dear authors of the article "Cell-death metabolites from Cocconeis scutellum var. parva identified by integrating bioactivity-based fractionation and non-targeted metabolomic approaches", your work deals with a highly interesting topic and I would like to congratulate you to this fine publication. I have no further comments.
Author Response
Dear Reviewer,
Thank you very much for your positive feedback on our article titled "Cell-death metabolites from Cocconeis scutellum var. parva identified by integrating bioactivity-based fractionation and non-targeted metabolomic approaches." We are delighted to hear that you found our work interesting and appreciate your congratulations on the publication. Your supportive comments mean a lot to us and encourage us to continue our research in this fascinating area.
Best regards,
Reviewer 2 Report
Comments and Suggestions for Authors
This study aims at identifying the metabolite(s) produced by a benthic diatom that is/are responsible for sex-reversal in a small crustacean. The authors nicely expose their experimental work that combines classical and novel approaches in chemical ecology. The manuscript is easy to read and the results well presented.
Before publication, I have a couple of comments that need to be addressed. My main interrogation is about the extraction procedure chosen by the authors. It is an unusual protocol, which has probably been selected based on preliminary assays. This should be stated somewhere in the manuscript along with an explanation on why the authors did not investigate the non polar extract and fraction although previous studies had shown volatile and non polar fractions of this diatom were active against the crustacean.
Please find below my other comments:
Methods
Line 297 : the culture conditions were similar to the first growth?
Line 275: what was the sorting? How did you choose the specimens?
Line 279-280: see units
Line 278: how the temperature was chosen? What was harvesting season?
Line 281-282: I am not aware of the life cycle of these shrimps, how did the authors anticipate they would release larvae? Are larvae visible with the naked eye? Are they easy to collect?
Line 283: what is “clean” filtered seawater? Was the seawater used for the experiments collected at the sampling site for shrimps and diatom?
Line 307-311: this is an unusual protocol for metabolites extraction, was it based on previous work? If so, it would be good to mention it. Indeed, with this protocol the authors miss non polar compounds that were mentioned in the introduction as potential candidates.
Line 323: since a non polar compound was expected, could the authors explain the rationale for chosing anion exchange chromatography, and overall this protocol? Did they expect these compounds to be charged?
Line 335: the reference to the section should be updated (refers to results)
Line 334-335: could the authors detail the rationale for going back to lipophilic extraction given that these extracts were obtained from lipophilic extraction initially?
Line 340: Is the column Grom-Sil 120 ODS-4 HE a C18 column? Could you please provide detail on the interaction phase?
Line 360: are the diatom extract being 2 or 1?
Line 362: refrigerator must be freezer
Line 363: the sentence is confusing, it seems like the food contained post larvae, please modify
Line 366: are fraction = extracts?
Line 367: how did the authors differentiate dead from alive post-larvae?
Line 371: maybe add a few details on how the pleopods were collected and further mounted on slides.
Line 377: update the section number
Line 381: see unit format
Line 385: why positive ionization was not performed? Were the authors only expecting a negatively charged ion? This aspect should be developed to give some explanation to the reader. (This is further mentioned in the results I believe)
Line 411: Are the authors comparing ion intensities within a same fraction? Higher intensities do not means higher amount; it all depends on their ionization pattern. Only a comparison between samples can be done for one ion. Please make sure this is clearly mentioned in the manuscript to avoid any confusion.
Given that peak picking was performed separately on each fraction, I am curious, were the retention times similar between runs? Comparing batches often requires additional steps for aligning the samples together
Line 436: I am not sure to understand what means “each growth phase”. Are the authors referring to each bioreactor? Or are they referring to exponential/stationary phase?
Line 447: I am not sure to understand what is the difference between line 447 and line 453?
Line 457: this sentence is not needed, this is obvious from the title and previous section
Line 458: this could be mentioned at the beginning of the section. But it also repeats a sentence at the beginning of the methods. The authors could remove it
Line 462: Rinsing the diatom biofilm with MilliQ water (pH around 6) could modify their metabolism, which would in turn induce a bias in the experiment. Same for freezing at -20°C, flash freezing is usually recommended to avoid modifications in the metabolism
Line 501: I am not sure how the authors did to select only one ion per metabolite. Are they saying that they excluded all isotopes, adducts and fragments for one single feature (ion and not metabolite)?
Line 510: reference to section should be updated (this is results)
Results
Line 93: what is “Plus” in Figure S1 A ? It should be mentioned in the caption.
Table 1 and 2: the authors should present 4 digits after coma for the m/z and 2 for retention times for all features.
Table 2: the second feature shows a potential adduct at 533.2884, was the adduct identified? It does not correspond to any classical adducts. Are the authors certain these ions are related to each other?
Line 154: it would be interesting to know the matching percentage for this putatively annotated feature
Discussion
Line 174 – 199: there is a lot of blabla in this section, the message could gain in clarity if the text was more concise. For instance, the two paragraphs on acidification could be combined to get straight to the message (up-regulated metabolites in non-acidified conditions).
Line 178-180: this sentence should be revisited, it is not surprising fraction numbers are different between studies, it all depends on your extraction/fractionation procedure. It is not necessary to emphasize on this aspect. It would rather be more interesting to dig a little bit on what has been done in the other study to investigate if metabolites of similar chromatographic behavior could be involved.
Line 183: the authors mentioned in the introduction that pH modification affects the bioactivity, not the metabolic profile. Furthermore, since the authors chose an extraction procedure targeting mostly polar non volatile compounds I am not sure to understand this justification.
Line 184-185: if the volatile profile is modified, this necessary comes from modification in the endometabolome. The whole paragraph should be revisited to make more sense
Line 217-223: since this is the authors’ main result, I would recommend placing this paragraph line 207 after the first discussion on LPG. In general, it would be best to have the discussion on LPG as a block and mention putative other fatty acids in another block. For fatty acids, you may want to refer to the various studies on chemical interactions between diatoms and copepods: Ponhert et al., 2005; Ianora et al., 2004 …
Line 250-253: it seems difficult to conclude with this sentence since the authors did not confirm the targeted bioactivity of LPG with a standard compound (as mentioned in the conclusion in the following small paragraph). The authors should be less affirmative.
Author Response
Dear Reviewer,
We appreciate your insightful comments and suggestions, which have greatly contributed to improving the clarity and quality of our manuscript. We have carefully considered each of your points and made several revisions to address your concerns.
Following you can find the answers to all your suggestions:
Materials and methods
Line 297: the culture conditions were similar to the first growth?
- Answer: It is well known that changes in culture conditions can significantly alter the production of bioactive molecules, sometimes dramatically (https://doi.org/10.1371/journal.pone.0218238). Therefore, we maintain consistent culture conditions for these diatoms to ensure that the diatoms produce the desired bioactive compounds. In addition, we know the growth curve of these species at a given temperature and irradiance and with a given medium. We have now rephrased the paragraph to clarify this question.
Line 275: what was the sorting? How did you choose the specimens?
- Answer: Specimens were sorted onboard to separate Hippolyte specimens from by-catch. This process was conducted using white containers and by well-trained operators. Fortunately, Hippolyte specimens can be easily identified in the Ischia meadows as no similar species are present, ensuring accurate sorting. Hippolyte specimens were then identified at the species level in the laboratory under a Leica Mesoscope. The paragraph has been extended with this information to make it more straightforward for the readers.
Line 279-280: see units
- Answer: The units have been corrected
Line 278: how the temperature was chosen? What was harvesting season?
- Answer: The harvest season was in spring. The temperature was chosen based on two main criteria: (a) the average seawater temperature at the field sites during this period and (b) a standardized rearing protocol that has been published in various papers (e.g., https://doi.org/10.1007/s00435-018-0405-z, doi:10.1017/S1751731117000908, https://doi.org/10.1038/s41598-019-48110-7). Given the significant influence of temperature on shrimp physiology, including its effects on nutrition, development, and apoptosis, it is not advisable to modify the temperature in a well-established protocol. We have now included the two references after the sentence.
Line 281-282: I am not aware of the life cycle of these shrimps, how did the authors anticipate they would release larvae? Are larvae visible with the naked eye? Are they easy to collect?
- Answer: Larvae can be easily identified under stereomicroscope. Furthermore, we know the reproductive cycle of this species, and it can be considered a standardized model organism for several scientific investigations. We added references where the readers can get more information regarding this procedure.
Line 283: what is "clean" filtered seawater? Was the seawater used for the experiments collected at the sampling site for shrimps and diatom?
- Answer: Our Laboratory, Ischia Marine Centre, is next to the sampling sites. We collect natural seawater. It is filtered at 200um and then sterilized. We have changed the text from "Clean filtered seawater" to "Filtered and sterilized natural seawater."
Line 307-311: this is an unusual protocol for metabolites extraction, was it based on previous work? If so, it would be good to mention it. Indeed, with this protocol the authors miss non polar compounds that were mentioned in the introduction as potential candidates.
- Answer: We agreed with the reviewer's comment. We now mention using a protocol established earlier by Zupo et al. (2014). It was shown that a polar extraction with 80 % acetonitrile yielded biological activity.
Line 323: since a non polar compound was expected, could the authors explain the rationale for chosing anion exchange chromatography, and overall this protocol? Did they expect these compounds to be charged?
- Answer: Since the activity could be extracted with 80% acetonitrile, we concluded that it is only moderately lipophilic, which allows us to speculate that it may be negatively or positively charged. We now give a rationale for our approach by saying in the paragraph: 'In order to find another property of the biologically active substance in addition to lipophilicity that can be used for chemical characterization, we tested for a negative charge of the substance using an anion-exchanger.'
Line 335: the reference to the section should be updated (refers to results)
- Answer: Updated, thanks
Line 334-335: could the authors detail the rationale for going back to lipophilic extraction given that these extracts were obtained from lipophilic extraction initially?
- We have now detailed in the paragraph: 'in order to re-extract the active compounds and thereby remove accompanying salts'.
Line 340: Is the column Grom-Sil 120 ODS-4 HE a C18 column? Could you please provide detail on the interaction phase?
- We have added the requested detail in the paragraph with: ‘C18-reversed-phase’
Line 360: are the diatom extract being 2 or 1?
- Answer: 2 parts of food and 1 part of diatoms. We have now change it in the paragraph to clarify this aspect: ‘For the positive controls, dried diatoms ( scutellum var. parva from identical batches) were added to the base food at 2:1 ratio (w/w) of base food:diatoms, respectively.’
Line 362: refrigerator must be freezer
- Answer: the reviewer is right. It is a freezer. It was corrected in the manuscript.
Line 363: the sentence is confusing, it seems like the food contained post larvae, please modify
- Answer: The sentence has been rephrased as follows: Daily, 5 mg of dry food, or dry food combined with diatoms, was administered to each vessel containing 25 inermis post-larvae, according to the specific treatment.
Line 366: are fraction = extracts?
- Answer: We have now clarified in this section that we used the dried fractions obtained in 4.3.2.4 instead of extracts.
Line 367: how did the authors differentiate dead from alive post-larvae?
- Answer: Determining mortality is a straightforward process. Dead post-larvae are pale white and do not move compared to the copper colour of alive and mobile larvae. Under a stereomicroscope, no internal activity can be observed. This information has been included in the paragraph.
Line 371: maybe add a few details on how the pleopods were collected and further mounted on slides.
- Answer: As the collection of pleopods is a well-established technique, we have added two references where the process is described to provide detailed information
Line 377: update the section number
- Answer: The section number has been updated. Thanks.
Line 381: see unit format
- Answer: The unit format has been corrected.
Line 385: why positive ionization was not performed? Were the authors only expecting a negatively charged ion? This aspect should be developed to give some explanation to the reader. (This is further mentioned in the results I believe)
- Answer: We measure only the negative ionization mode because the metabolites product of the strong anion-exchange solid phase extraction from section 4.3.2.3 must possess a negative charge nature. We now include this rationale in the paragraph.
Line 411: Are the authors comparing ion intensities within a same fraction? Higher intensities do not means higher amount; it all depends on their ionization pattern. Only a comparison between samples can be done for one ion. Please make sure this is clearly mentioned in the manuscript to avoid any confusion.
- Answer: We compared the expected changes in the intensities of the features among the various fractions (active and non-active) by defining a pattern that is a stated series of numbers, where each number represents the expected change of the features among the fractions. For example, the pattern "0-1-0-0-0-0-0-0-1" corresponded to the levels in "F1-F2-F3-F4-F5-F6-F7-F8-SAX" and was used to search for features more abundant in fraction F2 and SAX extract. We have rephrased this paragraph to avoid confusion.
Given that peak picking was performed separately on each fraction, I am curious, were the retention times similar between runs? Comparing batches often requires additional steps for aligning the samples together
- Answer: The fact that we initially sorted the converted files into separate folders according to the fraction numbers or SAX extract does not mean we performed the peak picking separately on each fraction. On the contrary, the centWave algorithm from XCMS 3.14.1 R package, perfom a peak picking among the separate folders (among fractions) and then peaks are aligned, the retention times are corrected with the obiwarp method, and then a second alignment is executed. As described in the manuscript, a bw=2 (Allowable retention time deviations in seconds) for the feature alignment was used to define a feature among the fractions. We have made a slight change that we consider help to avoid the confusion: ‘Converted files of all replicates were sorted into separate folders according to the fraction numbers or SAX extract, and then all folders were processed together with the XCMS 3.14.1 R package [55-57]
Line 436: I am not sure to understand what means "each growth phase". Are the authors referring to each bioreactor? Or are they referring to exponential/stationary phase?
- Answer: Some cells will break when the operator scraps the diatoms semi-massive cultures. For this reason, it is not possible to quantify the number of diatoms at time 0 of the culture curve. The number of cells in the semi-massive cultures or the number of cells in the medium we inoculate could not be considered accurate. We have rephrased with: ‘Due to the adhesive nature of Cocconeis spp., only a fraction of the diatoms inoculated into each photobioreactor survived each reinoculation process.’
Line 447: I am not sure to understand what is the difference between line 447 and line 453?
- Answer: The reviewer is right; there was a redundancy that has now been corrected. The paragraph has been rephrased to correct the redundancy and make it easier for the reader.
Line 457: this sentence is not needed, this is obvious from the title and previous section
- Answer: We agreed with the reviewer. This sentence has been deleted accordingly.
Line 458: this could be mentioned at the beginning of the section. But it also repeats a sentence at the beginning of the methods. The authors could remove it
- Answer: This sentence has been deleted accordingly.
Line 462: Rinsing the diatom biofilm with MilliQ water (pH around 6) could modify their metabolism, which would in turn induce a bias in the experiment. Same for freezing at -20°C, flash freezing is usually recommended to avoid modifications in the metabolism
- Answer: The washing procedure was necessary to eliminate residual salts that could interfere with further extractions and processes. We performed this washing for one second, and each empty photoreactor containing a tin layer of diatoms was frozen immediately to quench the metabolism and avoid the modifications mentioned by the reviewer. We used these techniques in previous papers and demonstrated the activity with Bioassays on inermis (https://doi.org/10.1371/journal.pone.0218238). We have now rephrased the sentence to clarify.
Line 501: I am not sure how the authors did to select only one ion per metabolite. Are they saying that they excluded all isotopes, adducts and fragments for one single feature (ion and not metabolite)?
- Answer: This is a common procedure in nontargeted metabolomics data analysis. For statistical computation, it is sometimes better to only work with putative compounds rather than all their adducts. Features generated by the XCMS R package were initially annotated using the CAMERA R package. Within the CAMERA R Package exists the method to generate a reduced peak list from the annotated peak list using the function “getReducedPeaklist”. This function merges all adducts of a potential compound into a single representative compound within each pcgroup . As a result, the simplified peak list includes only one annotated feature per group. Thus, the reduced peak list only contains one annotated feature per group. Once interesting features were selected after statistics. The data is cross-checked, including all the features belonging to one pcgroup. The paragraph has been extended with this information to make it clear for the readers.
Line 510: reference to section should be updated (this is results)
- Answer: The section has been updated
Results
Line 93: what is "Plus" in Figure S1 A? It should be mentioned in the caption.
- Answer: We have now mentioned in the caption of Figure S1A that ‘Plus’ means 'positive control'. What is behind 'positive control' is explained in section 4.3.3
Table 1 and 2: the authors should present 4 digits after coma for the m/z and 2 for retention times for all features.
- Answer: The digits have been corrected for all the m/z and retention time values
Table 2: the second feature shows a potential adduct at 533.2884, was the adduct identified? It does not correspond to any classical adducts. Are the authors certain these ions are related to each other?
- Answer: Unfortunately, there was a typographical error in the previous version, which has now been corrected from 533.2884 to 544.2884. This m/z was annotated as a [M+K-2H]- adduct
Line 154: it would be interesting to know the matching percentage for this putatively annotated feature
- The SIRIUS outcome: Sirius Score: 91.078%, Total explained intensity: 90.628%. Now has been included in the paragraph for the readers.
Discussion
Line 174 – 199: there is a lot of blabla in this section, the message could gain in clarity if the text was more concise. For instance, the two paragraphs on acidification could be combined to get straight to the message (up-regulated metabolites in non-acidified conditions).
- Answer: Thank you for your insightful feedback. We acknowledge that the section contains extensive details which could be streamlined for better clarity. We have combined the two paragraphs on acidification to provide a more direct and concise message regarding the up-regulated metabolites in non-acidified conditions. We hope this revision enhances the readability of our findings.
Line 178-180: this sentence should be revisited, it is not surprising fraction numbers are different between studies, it all depends on your extraction/fractionation procedure. It is not necessary to emphasise on this aspect. It would rather be more interesting to dig a little bit on what has been done in the other study to investigate if metabolites of similar chromatographic behavior could be involved.
- Answer: Thank you for your constructive comment. We agree that emphasising the differences in fraction numbers between studies is unnecessary. Instead, we will focus on the methodologies used in the other study to discuss if metabolites with similar chromatographic behaviours are involved. We believe this approach will provide more valuable insights and strengthen our discussion. The paragraph has been modified, including the suggestion from the reviewer.
Line 183: the authors mentioned in the introduction that pH modification affects the bioactivity, not the metabolic profile. Furthermore, since the authors chose an extraction procedure targeting mostly polar non-volatile compounds, I am not sure I understand this justification.
- Answer: In lines 63-64 in the introduction of the reviewer's revised version, we mentioned: ‘Other studies have shown alterations in the exometabolome of scutellum under acidified conditions, affecting recognition by H. inermis [16]. With that evidence, and considering that changes in the exometabolome could be a product endometabolic changes, we decided to hypothesize that ocean acidification could influence also the endometabolome of C. scutellum parva.
Line 184-185: if the volatile profile is modified, this necessary comes from modification in the endometabolome. The whole paragraph should be revisited to make more sense
- Answer: The reviewer is correct in noting that modifications in the volatile profile typically indicate changes in the endometabolome. However, it is also true that not every endometabolic change results in altered volatile emissions. Our study aims to identify and understand endometabolic changes comprehensively, recognising that these changes may or may not manifest as differences in volatile emissions. This approach ensures a more holistic understanding of the endometabolome and its potential impacts beyond just volatile emissions. We have now rephrased and modified the paragraph to avoid confusion.
Line 217-223: since this is the authors' main result, I would recommend placing this paragraph line 207 after the first discussion on LPG. In general, it would be best to have the discussion on LPG as a block and mention putative other fatty acids in another block. For fatty acids, you may want to refer to the various studies on chemical interactions between diatoms and copepods: Ponhert et al., 2005; Ianora et al., 2004 …
- Answer: We have now reorganised the discussion paragraphs into two separate blocks to facilitate the flow of the discussion. The two studies suggested by the reviewer have also been incorporated into the discussion.
Line 250-253: it seems difficult to conclude with this sentence since the authors did not confirm the targeted bioactivity of LPG with a standard compound (as mentioned in the conclusion in the following small paragraph). The authors should be less affirmative.
- Answer: We have now made the conclusion less affirmative.
Reviewer 3 Report
Comments and Suggestions for Authors
The article presents a bioactivity-based fractionation and a metabolomic approach to identify potential apoptotic metabolites of Cocconeis scutellum var. The article presents exciting results since there is an intriguing aspect of some benthic diatoms: the production of an unidentified compound that promotes cell death. Thus, the article needs to answer the following questions:
1) What is the authors' conclusion regarding the chemical findings obtained by LC-MS?
2) Was it possible through this study to find the supposed identity of the unidentified compound that promotes cell death?
Author Response
Dear Reviewer,
We appreciate your insightful comments and suggestions, which have greatly contributed to improving the clarity and quality of our manuscript. We have now included a more robust final conclusive paragraph in the discussion section to address the questions you raised. Specifically:
-
What is the authors' conclusion regarding the chemical findings obtained by LC-MS?
We have elaborated on our conclusions regarding the chemical findings obtained by LC-MS in the new conclusive paragraph. Our analysis revealed several potential apoptotic metabolites, and we have discussed the implications of these findings in detail.
-
Was it possible through this study to find the supposed identity of the unidentified compound that promotes cell death?
We have addressed this question in the revised discussion. While our study has provided significant insights, the exact identity of the unidentified compound that promotes cell death remains elusive. However, we have identified several candidate metabolites that warrant further investigation, and we have discussed the potential pathways and mechanisms involved.
Thank you once again for your valuable feedback. We believe these revisions enhance the manuscript and address your concerns comprehensively.
Round 2
Reviewer 3 Report
Comments and Suggestions for Authors
The authors adequately answered the questions